# *Coffea arabica* Extracts and Metabolites with Potential Inhibitory Activity of the Major Enzymes in *Bothrops asper* Venom

**DOI:** 10.3390/ph18081151

**Published:** 2025-08-01

**Authors:** Erika Páez, Yeisson Galvis-Pérez, Jaime Andrés Pereañez, Lina María Preciado, Isabel Cristina Henao-Castañeda

**Affiliations:** 1Toxinology, Therapeutic, and Food Alternatives, Faculty of Pharmaceutical and Food Sciences, University of Antioquia, Medellin 050010, Colombia; erika.paez@udea.edu.co (E.P.); yeisson.galvis@udea.edu.co (Y.G.-P.); 2Research Group on Pharmaceutical Promotion and Prevention, University of Antioquia, Medellin 050010, Colombia; jaime.pereanez@udea.edu.co; 3Research Group on Bioactive Substances (GISB), Faculty of Pharmaceutical and Food Sciences, University of Antioquia, Medellin 050010, Colombia; 4Research Group on Marine Natural Products, Faculty of Pharmaceutical and Food Sciences, University of Antioquia, Medellin 050010, Colombia

**Keywords:** *Bothrops asper*, chlorogenic acid, coffee, molecular docking simulation, snake venoms

## Abstract

**Background/Objectives**: Most snakebite incidents in Latin America are caused by species of the *Bothrops* genus. Their venom induces severe local effects, against which antivenom therapy has limited efficacy. Metabolites derived from *Coffea arabica* have demonstrated anti-inflammatory and anticoagulant properties, suggesting their potential as therapeutic agents to inhibit the local effects induced by *B. asper* venom. **Methods**: Three enzymatic assays were performed: inhibition of the procoagulant and amidolytic activities of snake venom serine proteinases (SVSPs); inhibition of the proteolytic activity of snake venom metalloproteinases (SVMPs); and inhibition of the catalytic activity of snake venom phospholipases A_2_ (PLA_2s_). Additionally, molecular docking studies were conducted to propose potential inhibitory mechanisms of the metabolites chlorogenic acid, caffeine, and caffeic acid. **Results**: Green and roasted coffee extracts partially inhibited the enzymatic activity of SVSPs and SVMPs. Notably, the green coffee extract, at a 1:20 ratio, effectively inhibited PLA_2_ activity. Among the individual metabolites tested, partial inhibition of SVSP and PLA_2_ activities was observed, whereas no significant inhibition of SVMP proteolytic activity was detected. Chlorogenic acid was the most effective metabolite, significantly prolonging plasma coagulation time and achieving up to 82% inhibition at a concentration of 62.5 μM. Molecular docking analysis revealed interactions between chlorogenic acid and key active site residues of SVSP and PLA_2_ enzymes from *B. asper* venom. **Conclusions**: The roasted coffee extract demonstrated the highest inhibitory effect on venom toxins, potentially due to the formation of bioactive compounds during the Maillard reaction. Molecular modeling suggests that the tested inhibitors may bind to and occupy the substrate-binding clefts of the target enzymes. These findings support further in vivo research to explore the use of plant-derived polyphenols as adjuvant therapies in the treatment of snakebite envenoming.

## 1. Introduction

Snakebite envenoming is a neglected tropical disease resulting from the injection of a highly specialized toxic secretion into humans. It primarily occurs in rural areas, with more than five million cases reported annually, leading to between 81,000 and 138,000 deaths. However, this number may triple when accounting for amputations and disabling complications [1]. Most cases are concentrated in Asia, Africa, and Latin America [2]. In 2023, 5421 snakebite incidents were reported in Colombia (117 cases per week), including accidents involving both venomous and non-venomous snakes [3].

In Central America and northern South America, most snakebite incidents are caused by snakes of the *Bothrops* genus (subfamily Crotalinae, family Viperidae), which are distributed from southern Mexico to Argentina. *B. asper* is responsible for 50–80% of these cases [4]. Envenoming by *B. asper* produces both local and systemic symptoms, including pain, edema, dermonecrosis, myotoxicity, defibrination, thrombocytopenia, local and systemic hemorrhage, hypotension, and acute renal failure. Additional complications may include soft tissue infections leading to limb amputation, compartment syndrome, central nervous system hemorrhage, acute renal failure, and premature placental abruption [5].

Snake venoms are composed of biologically active molecules, with nearly 90% of their dry weight consisting of protein components responsible for local tissue damage and systemic toxicity, as well as relevant for treatment approaches [1]. The most abundant components of *Bothrops asper* venom in Colombia include snake venom serine proteinases (SVSPs), snake venom metalloproteinases (SVMPs), and snake venom phospholipases A_2_ (SVPLAs) [6].

The main therapeutic strategy for snakebite envenoming is the intravenous administration of antivenom, which consists of purified immunoglobulin G obtained from animal plasma. However, several limitations affect the efficacy and accessibility of antivenoms. These include limited availability in rural areas, where most bites occur, requirements for administration by trained healthcare professionals due to the risk of adverse reactions such as anaphylaxis, and their limited efficacy in neutralizing local tissue damage caused by snake venom [7]. Consequently, there is a need to investigate new inhibitory molecules that can complement antivenom therapy. Small-molecule toxin inhibitors are considered promising candidates for developing broad-spectrum snakebite treatments, as they can neutralize venom enzymatic activities [8].

Beyond being one of the most widely consumed stimulating beverages worldwide, coffee has demonstrated important bioactive properties, which have been the focus of research for their pharmacological role in the prevention and treatment of chronic non-communicable diseases, including cancer, cardiovascular diseases, Parkinson’s disease, and Alzheimer’s disease. *Coffea arabica* has also been traditionally used in folk medicine. A review by [9] compiled traditional uses of coffee beans and other plant parts. In Haiti, leaves and roasted beans are used to treat anemia, edema, and asthenia. In Nicaragua, a bean decoction is used orally to treat fever; in Peru, an aqueous extract of dried beans is used for fatigue; in Cuba, a hot water extract of beans is considered an aphrodisiac; and in Brazil, beverages prepared from *C. arabica* beans are used to treat flu. Additionally, roasted beans are used in Africa to treat headaches, migraines, malaria, and general weakness [9].

Various biological activities have been demonstrated for *C. arabica* seed extracts. For example, methanolic extracts have shown the ability to prevent skin photodamage by inhibiting the expression of metalloproteinases-1, -3, and -9, as well as the MAP kinase pathway [10]. *C. arabica* seed extracts and their main phenolic compounds, such as caffeic and chlorogenic acids, also exhibit antinociceptive and anti-inflammatory effects in a murine gout model, reducing hypernociception, neutrophil migration, and the concentration of pro-inflammatory cytokines [11]. Moreover, anticancer activity has been reported, such as a dose- and time-dependent reduction in the viability of colorectal cancer cell lines [12].

Therefore, this study aims to evaluate the inhibitory potential of green (GCE) and roasted (RCE) coffee bean extracts, as well as major metabolites found in *Coffea arabica* (including chlorogenic acid, caffeine, and caffeic acid), cultivated in Colombia, against the coagulant, proteolytic, and phospholipase A_2_ activities of *Bothrops asper* venom. Both GCE and RCE were analyzed, as it has been demonstrated that the roasting process alters the chemical composition and biological activity of coffee: while some natural phenolic compounds may be lost, other antioxidant compounds, such as Maillard reaction products, are formed [13]. Additionally, this study employs molecular docking techniques to propose potential inhibitory mechanisms between the evaluated secondary metabolites and key venom enzymes from *B. asper*.

## 2. Results

### 2.1. Characterization of Coffea arabica Extracts

The extracts were characterized by HPLC-DAD, which revealed the presence of caffeine at concentrations of 28.8 mg/g (GCE) and 33.7 mg/g (RCE). At the same time, chlorogenic acids (comprising chlorogenic, neochlorogenic, and cryptochlorogenic acids) were 208.55 mg/g (GCE) and 75.78 mg/g (RCE). Consider that caffeoylquinic acids are a bound form of caffeic acid present in green and roasted coffee at high levels [14]. This study also includes caffeic acid as a relevant metabolite.

### 2.2. Inhibition of Pro-Coagulant Activity of *Bothrops asper* Venom 

None of the tested concentrations of extracts or individual metabolites exhibited intrinsic coagulant activity when added to citrated human plasma. In contrast, the positive control (venom alone) induced coagulation within 20 to 30 s. However, coagulant activity was significantly inhibited by both green coffee extract (GCE) and roasted coffee extract (RCE) at weight/weight (*w*/*w*) ratios of 1:20 and 1:40. For GCE, the coagulation times increased to 42.61 ± 0.76 s and 48.46 ± 1.56 s (*p* < 0.05), while RCE yielded coagulation times of 41.43 ± 3.69 s and 51.58 ± 5.76 s (*p* < 0.05), respectively (Figure 1A).

Chlorogenic acid showed the greatest inhibitory effect, prolonging the plasma coagulation time by over five minutes at concentrations between 125 and 500 μM (*p* < 0.05). Additionally, it partially inhibited *B. asper* venom-induced coagulant activity at lower concentrations of 31.5 and 62.5 μM (*p* < 0.05). Caffeic acid also demonstrated a dose-dependent inhibitory effect, with significant differences compared to the positive control observed at concentrations from 62.5 to 500 μM (*p* < 0.05). In contrast, caffeine only exhibited inhibitory activity at the highest concentration tested (Figure 1B). The effect size for chlorogenic acid, calculated using Cohen’s d, was 4.09, indicating a strong inhibitory effect (Appendix A).

### 2.3. Inhibition of Amidolytic Activity

The inhibition of the amidolytic activity of *Bothrops asper* venom was evaluated using the colorimetric substrate BApNA (Nα-Benzoyl-DL-arginine 4-nitroanilide), a standard probe for serine proteases. Both extracts demonstrated significant inhibitory effects against the venom amidolytic enzymes at weight-to-weight (*w*/*w*) ratios of 1:5, 1:10, and 1:20 (*p* < 0.05), compared to the positive control (Figure 2A). Notably, Cohen’s d value for roasted coffee extract was 6.06, indicating a strong effect size (Appendix A).

Caffeine and caffeic acid partially inhibited this venom activity at concentrations of 125–500 μM (*p* < 0.05). Chlorogenic acid only displayed a slight inhibition at 500 μM (*p* < 0.05) (Figure 2B).

### 2.4. Inhibition of PLA_2_ Activity

Phospholipase A_2_ (PLA_2_) activity of *B. asper* venom was assessed using the 4-NOBA (4-Nitro-3-octanoyloxy benzoic acid) substrate. Green coffee extract (GCE) showed partial inhibition at a *w*/*w* ratio of 1:20, while the other tested ratios (1:1, 1:5, 1:10) and all concentrations of roasted coffee extract (RCE) had no significant inhibitory effect (Figure 3A). All the tested metabolites demonstrated significant inhibition at 500 μM (*p* < 0.05), with activity reduced by approximately 50%. Additionally, chlorogenic acid partially inhibited 4-NOBA hydrolysis at 250 μM (Figure 3B).

### 2.5. Inhibition of Proteolytic Activity

Both extracts also partially inhibited the proteolytic activity of *B. asper* venom on azocasein at all tested w/w ratios (*p* < 0.05). The highest inhibitory effect was observed with RCE at a ratio of 1:20, resulting in approximately 50% reduction of proteolytic activity (Figure 4A). In contrast, none of the tested metabolites showed significant inhibition of this enzymatic activity (*p* > 0.05) (Figure 4B).

### 2.6. Structural Modeling and Validation of the SVSP

Molecular modeling generated a three-dimensional structure exhibiting the characteristic features of snake venom serine proteinases (SVSPs), including two β-barrels and four short α-helices, consistent with a trypsin-like fold (Figure 5A). The structural model was validated using PROCHECK (available on https://saves.mbi.ucla.edu/, accessed on 25 March 2025), which revealed that 91.5% of residues were located in the most favored regions of the Ramachandran plot, and 10.2% were in additional allowed regions. No residues were found in generously allowed or disallowed regions (Appendix A). The Verify3D analysis showed that 87.83% of the residues scored ≥ 0.1, surpassing the 80% threshold for reliable models (Appendix A). The ProSA analysis returned a Z-score of −7.26, within the range expected for proteins of comparable length. Additionally, the energy profile of the model indicated a predominance of residues in favorable (negative) energy regions (Appendix A).

### 2.7. Molecular Docking

Molecular docking revealed the interactions of the studied metabolites with the modeled SVSP of *B. asper* (Figure 5). The binding affinities and interaction details are summarized in Table 1. Chlorogenic acid exhibited the strongest predicted interaction, forming hydrogen bonds with Asn41 and Ser181, residues located in the enzyme active site (Figure 5C). Caffeic acid formed fewer hydrogen bonds and did not interact with catalytic residues (Figure 5B). Caffeine showed the weakest affinity, forming only one hydrogen bond with Thr176 (Figure 5D).

On the other hand, the molecular docking results for *Bothrops asper* PLA_2_ are presented in Figure 6. The tested compounds interacted with amino acid residues located within the enzyme catalytic site or hydrophobic channel. The binding affinities and detailed interaction profiles are summarized in Table 2. Among the metabolites, chlorogenic acid exhibited the most favorable theoretical binding affinity, forming interactions with two key catalytic residues: a salt bridge with His48 and a combination of a salt bridge and Van der Waals interactions with Tyr52 (Figure 6B). In contrast, caffeic acid and caffeine showed equivalent binding affinities. Caffeic acid established a salt bridge with His48, along with additional weak interactions with surrounding residues (Figure 6A), whereas caffeine formed π-stacking interactions with Phe5, a residue located within the hydrophobic channel (Figure 6C).

## 3. Discussion

*B. asper* is the most clinically relevant species in Colombia and other regions. Its venom is mainly composed of proteins and peptides [15], with the most abundant components being enzymes from the groups PLA_2_s, SVMPs, and SVSPs [6]. Therefore, green and roasted coffee bean extracts from *Coffea arabica*, along with some of their constituent metabolites, were evaluated for their ability to inhibit the enzymatic and biological activities induced by these toxins.

This study constitutes the first report demonstrating the in vitro inhibitory capacity of green and roasted *C. arabica* beans against snake-venom-induced biological activities. *C. arabica* is well known for its richness in bioactive metabolites, including chlorogenic, caffeic, and ferulic acids and catechin, epicatechin, and anthocyanins, as well as alkaloids, such as caffeine and trigonelline. During roasting, additional compounds are formed, such as melanoidins and acrylamide, which may also contribute to its biological activity [16].

*C. arabica* has exhibited various pharmacological properties, including anti-inflammatory and analgesic effects, which may offer therapeutic potential for snakebite treatment [11]. These findings, together with those presented herein, underscore the potential of this plant as a promising source for the identification of antiophidic compounds.

The proteolytic activity of snake venom is closely associated with its hemorrhagic potential [17]. Accordingly, inhibition of proteolytic enzymes is correlated with a reduction in hemorrhagic effects. In the present study, *C. arabica* extracts partially inhibited the proteolytic activity of *B. asper* venom. However, none of the individual compounds tested reduced venom activity at any of the concentrations evaluated. This suggests that the inhibitory effects observed for the extracts may be mediated by other untested molecules or through synergistic interactions among the components. Further studies are needed to explore these possibilities and identify the specific constituents responsible for the observed activity.

On the other hand, PLA_2_ enzymes in snake venoms contribute to myotoxicity, edema formation, and neurotoxicity, among other effects [18]. In this study, green coffee extract (GCE) exhibited mild inhibitory activity against PLA_2_, whereas the roasted coffee extract (RCE) showed no such effect. Interestingly, all the pure compounds tested inhibited PLA_2_ activity at the highest concentration used. The polyphenols and flavonoids present in green coffee are likely responsible for this activity [19]. However, the roasting process alters the composition of these compounds due to the formation of new molecules, such as pyrazines, pyrroles, thiols, and furanones, via the Maillard reaction, potentially affecting bioactivity [20]. Chlorogenic acid, a phenolic ester of cinnamic acid, is considered the main bioactive compound in green coffee beans [21]. In this study, chlorogenic acid inhibited PLA_2_ activity at concentrations of 250 and 500 µM, resulting in approximately a 50% reduction in activity. Similar results have been reported, including the inhibition of PLA_2_ and myotoxic activity of *Bothrops leucurus* venom [22], the neurotoxicity of *Crotalus durissus terrificus* venom [23], and the myotoxicity of a PLA_2_ homologue from *Bothrops jararacussu* venom [24].

Our molecular docking analysis suggests that chlorogenic acid may interact with the catalytic residue His48, which is essential for generating the hydroxyl group that attacks the carbonyl at the sn-2 position of the glycerophospholipid substrate [25]. This interaction could disrupt the enzyme’s catalytic cycle and inhibit its function. Additionally, chlorogenic acid may bind to Lys69, a residue implicated in interfacial recognition and stabilization of substrate binding [26], potentially blocking substrate access. These findings provide a structural basis for the observed inhibitory activity of chlorogenic acid against PLA_2_ enzymes in snake venom.

The in vitro procoagulant activity of *B. asper* venom is associated with its in vivo anticoagulant effects, primarily due to the consumption of fibrin and fibrinogen [27]. This effect is largely attributed to SVSPs and, to a lesser extent, SVMPs. These enzymes act on various substrates, including coagulation factors and other proteins [28,29]. In this study, both green and roasted *C. arabica* extracts partially inhibited the amidolytic, proteolytic, and coagulant activities of *B. asper* venom. The amidolytic activity is mediated by SVSPs acting on the synthetic substrate BApNA, which is cleaved by proteases with a preference for arginine at the P1 position [30]. Nevertheless, as previously noted, inhibition of amidolytic activity does not always correlate with inhibition of coagulant activity. This was observed in our study, as the concentrations of compounds that prolonged the coagulation time did not coincide with those that inhibited the hydrolytic activity on BApNA. Chlorogenic acid showed the strongest inhibitory effect, significantly increasing coagulation times (by 3- to 15-fold) even at low concentrations (31.6 µM). Similar results have been reported [31], including the chlorogenic-acid-induced inhibition of thrombin, factor V, and factor X. These findings motivated further molecular modeling to investigate the mechanism of action.

Docking simulations suggested that chlorogenic acid may bind to the active site of a thrombin-like SVSP, occupying part of the substrate-binding cleft. The inhibitor potentially forms hydrogen bonds with the catalytic Ser181, which mediates the nucleophilic attack on peptide bonds [32]. This interaction may impair the enzyme catalytic cycle. Additionally, the quinic acid moiety of chlorogenic acid may occupy the S1 subsite, interacting via hydrogen bonding with Thr176 and Trp197, and engaging in van der Waals interactions with Val195 and Trp197. Similar interactions have been reported for chlorogenic acid binding to thrombin [33,34], supporting its capacity to block substrate access and inhibit enzyme activity [32].

This manuscript acknowledges limitations inherent to experimental designs, such as sample size and random sampling, which may affect the study external validity. Moreover, the absence of confounder control and multivariate analysis may limit internal validity. Future in vivo studies are recommended to confirm the therapeutic potential of the studied metabolites.

## 4. Materials and Methods

### 4.1. Materials and Chemicals

Venom from *Bothrops asper* was obtained from adult specimens collected in Magdalena Medio, Urabá, the Pacific Coast, and Chocó (Colombia), and was provided by the Serpentarium of the University of Antioquia. A pooled sample was centrifuged, lyophilized, and stored at −20 °C until use. Solvent removal was conducted using a rotary evaporator, followed by lyophilization.

Green (unroasted) and medium-roasted coffee (*Coffea arabica*) samples (500 g each) were supplied by Natucafé S.A.S. (Andes, Antioquia, Colombia). Each sample was independently pulverized and extracted for 48 h with a mixture of isopropanol and water (60:40, *v*/*v*) at room temperature using a solid-to-liquid ratio of 1:3 (*w*/*v*). The extraction included sonication for 30 min at 0, 24, and 48 h. The resulting extract was vacuum filtered, concentrated via rotary evaporation at 40 °C, lyophilized, and stored at −20 °C.

Both extracts were characterized by HPLC-DAD. For this, 100 mg of lyophilized extract were dissolved in 70% ethanol and centrifuged at 13,000 rpm for 10 min, and the supernatant was collected and diluted to 20 mL in a volumetric flask. The final solution was further diluted 1:20 with the mobile phase and injected into an Agilent 1200 Series LC System equipped with an SB-C18 column [35]. Quantification was performed using external standard methodologies [12]. Reference standards—chlorogenic acid (≥95%), caffeine (≥99%), and caffeic acid (≥98%)—were purchased from Sigma-Aldrich (St. Louis, MO, USA) and dissolved in dimethyl sulfoxide (DMSO).

### 4.2. Inhibition of Procoagulant Activity of SVSPs

The procoagulant activity assay was performed based on the method described by [36], with minor modifications. The minimum coagulant dose (MCD), defined as the amount of venom that clots citrated human plasma within 20–30 s, was used as a positive control and diluted 1:1 in saline solution. Green coffee extract (GCE) and roasted coffee extract (RCE) were prepared at MCD/extract ratios of 1:1, 1:5, 1:10, 1:20, and 1:40. Chlorogenic acid, caffeine, and caffeic acid were tested at concentrations of 500, 250, 125, 62.5, 31.2, and 15.6 µM. All the treatments were adjusted to 500 µL with saline solution and preincubated at 37 °C for 30 min. Subsequently, 100 µL of each mixture were added to 200 µL of citrated human plasma in quadruplicate. Negative controls consisted of extracts or metabolites with DMSO at the highest evaluated concentration. The coagulation time was monitored for up to 5 min.

### 4.3. Inhibition of Amidolytic Activity of SVSPs

Amidolytic activity was evaluated using BApNA (*N*-Succinyl-arginine-*p*-nitroanilide; Sigma-Aldrich) as the substrate, following the methodology described in [37]. The reaction mixture included 50 µL of Tris-HCl buffer (10 mM Tris-HCl, 10 mM CaCl_2_, 100 mM NaCl; pH 8.0), 200 µL of BApNA (10 mM), and 10 µL of *B. asper* venom (2 mg/mL), preincubated with the extracts or metabolites. Extracts were tested at venom/extract ratios of 1:1, 1:5, 1:10, and 1:20, while metabolites were evaluated at the concentrations mentioned above. After 40 min incubation at 37 °C, the absorbance was measured at 415 nm. Positive controls contained venom and substrate only; negative controls included extract or metabolite samples without venom at all the evaluated concentrations.

### 4.4. Inhibition of Proteolytic Activity of SVMPs

The inhibition of proteolytic activity was assessed using azocasein as the substrate [38]. The minimum proteolytic dose (MPD), defined as the amount of venom producing a 0.5 absorbance increase at 450 nm, was determined as 20 µg of venom. This dose was diluted in 25 mM Tris buffer (150 mM NaCl, 5 mM CaCl_2_, pH 7.4). GCE, RCE, and the metabolites were tested at the same ratios and concentrations as previously described. After 30 min of preincubation at 37 °C, 100 µL of azocasein (10 mg/mL) were added and incubated for 90 min at 37 °C. The reaction was terminated with 200 µL of 5% trichloroacetic acid, centrifuged at 3000 rpm for 5 min, and 100 µL of the supernatant were mixed with 100 µL of 0.5 M NaOH. The absorbance was read at 450 nm. All the samples were assayed in quadruplicate. As a positive control, an MPD was used, and as a negative control, the extracts and metabolites were evaluated at all concentrations.

### 4.5. Inhibition of Esterase Activity of SVPLA_2_s

Esterase activity of phospholipase A_2_ (PLA_2_) was assessed using 4-NOBA as a substrate [39], adapted for 96-well plates. Each reaction contained 200 µL of buffer (10 mM Tris-HCl, 10 mM CaCl_2_, 100 mM NaCl, pH 8.0), 20 µL of 4-NOBA (1 mg/mL), and 20 µL of extract or metabolite solution preincubated with 20 µg of *B. asper* venom. Reactions were incubated at 37 °C for 60 min, and absorbance was measured at 405 nm using a plate reader (Awareness, Stat Fax 3200, Westport, CT, USA). All the treatments were tested in triplicate. *B. asper* venom and synthetic substrate 4-NOBA were positive controls. The extracts and metabolites, with DMSO at the highest evaluated concentration, were used as a negative control.

### 4.6. Molecular Modeling and Docking

Molecular structures of chlorogenic acid, caffeic acid, and caffeine were constructed using GaussView 6 [40]. Geometry optimization was performed using Gaussian 16W [41] at the B3LYP/6-31++G(d,p) level of theory. Molecular docking simulations were conducted using AutoDock Vina 1.2.0 [42] against two targets: a modeled thrombin-like serine protease from *B. asper* venom and the crystal structure of a PLA_2_ from *B. asper* venom (PDB ID: 5TFV), since the evaluated extracts or metabolites showed inhibition upon these toxins.

The SVSP was modeled based on the UniProtKB sequence Q072L6 (VSPL_BOTAS), with the signal and propeptide sequences removed. Structure prediction was carried out using AlphaFold 3 [43]. Further, the stereochemical quality of the modeled protein structure and overall structural geometry were confirmed using Procheck (available on https://saves.mbi.ucla.edu/, accessed on 25 March 2025) [44]. The energy of the residues was determined using ProSA through the ProSA-web service (available on https://prosa.services.came.sbg.ac.at/prosa.php, accessed on 25 March 2025). Finally, Verify 3D software (available on https://saves.mbi.ucla.edu/, accessed on 25 March 2025) was used to determine the compatibility of an atomic model (3D) with its amino acid sequence (1D) by assigning a structural class based on its location and environment (α, β, loop, polar, non-polar, etc.), as well as by comparing the results with suitable database structures.

Proteins were prepared using Maestro 2025-2 with the Protein Preparation Wizard. Hydrogen atoms were added, atomic charges were assigned, and local minimization was applied using the OPLS force field. The grid box was centered on the carbonyl oxygen of Ser181 for the SVSP (X = −3.079, Y = −6.399, Z = 1.810) and the calcium ion for PLA_2_ (X = 2.279, Y = 15.924, Z = 21.732), with a grid size of 24 Å^3^ and exhaustiveness set to 20. The ligand poses with the highest binding affinities were selected and analyzed using PLIP 2021 (available on https://plip-tool.biotec.tu-dresden.de/plip-web/plip/index, accessed on 28 March 2025). [45] and UCSF Chimera 1.19 [46].

### 4.7. Statistical Analysis

Statistical differences between the treatment groups and positive controls were assessed using two-way ANOVA, followed by Bonferroni post hoc tests. Differences were considered statistically significant at *p* < 0.05. Effect sizes were calculated using Cohen’s d, and all analyses were performed using Jamovi software 2.7.3.

## 5. Conclusions

This study evaluated the inhibitory potential of *Coffea arabica* extracts and their metabolites—chlorogenic acid, caffeic acid, and caffeine—against key enzymatic activities of *Bothrops asper* venom from Colombia. Through a combination of in vitro biochemical assays and molecular modeling, we demonstrated that both green coffee extract (GCE) and roasted coffee extract (RCE) exhibit significant inhibitory effects on serine proteases (SVSPs) and metalloproteinases (SVMPs), which are critical components of venom pathophysiological actions.

Among the individual metabolites, chlorogenic acid exhibited the most robust inhibition, particularly of procoagulant and PLA_2_ activities, as confirmed by prolonged clotting times. Molecular docking simulations further supported these findings, revealing favorable binding energies and multiple interactions between chlorogenic acid and the catalytic residues of both venom serine protease and PLA_2_. Caffeic acid also showed moderate inhibition, while caffeine exhibited limited activity.

These results underscore the therapeutic potential of chlorogenic acid and coffee extracts as natural inhibitors of snake venom enzymes. Given their availability, low toxicity, and dual antioxidant and enzyme-inhibitory properties, these compounds represent promising candidates for adjunctive treatment in snakebite envenoming. Future work should include in vivo efficacy testing and pharmacokinetic profiling.

## Figures and Tables

**Figure 1 pharmaceuticals-18-01151-f001:**
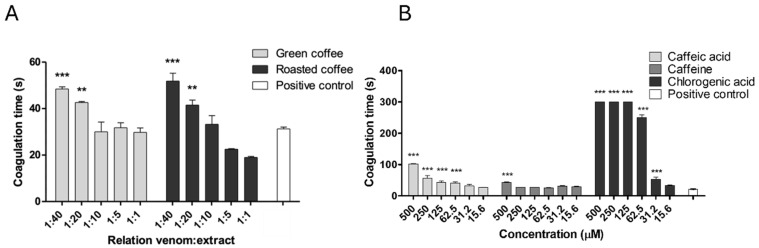
Inhibition of coagulant activity by *C. arabica* extracts and metabolites. (**A**) Green and roasted coffee extracts. (**B**) Caffeine, caffeic, and chlorogenic acid. Data are represented as the mean of coagulation time ± SEM (*n* = 4). *** represents statistically significant differences with the positive control with *p* < 0.001 and ** *p* < 0.01.

**Figure 2 pharmaceuticals-18-01151-f002:**
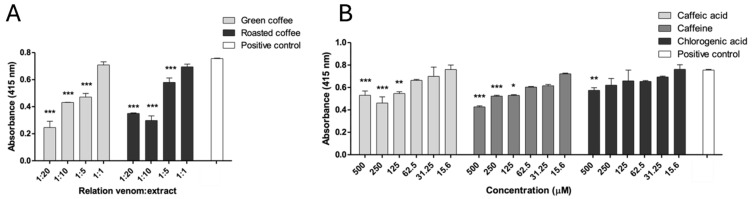
Inhibition of amidolytic activity of *B. asper* venom by *C. arabica* extracts and metabolites. (**A**) Green and roasted coffee extracts. (**B**) Caffeine, caffeic, and chlorogenic acid. Data are represented as mean ± SEM (*n* = 3). *** represents statistically significant differences with the positive control with *p* < 0.001, ** *p* < 0.01, and * *p* < 0.05. Data are represented as mean ± SEM (*n* = 3), with the subtraction of the negative control to delete background noise and the intrinsic compound absorbance at 415 nm.

**Figure 3 pharmaceuticals-18-01151-f003:**
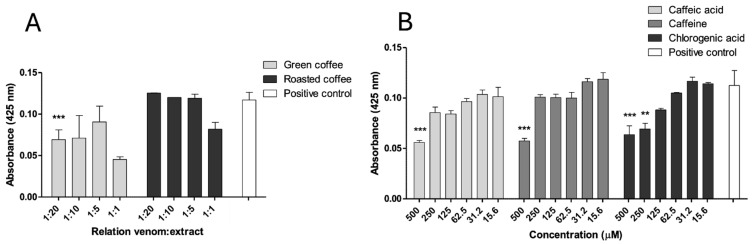
Inhibition of PLA_2_ activity of *B. asper* venom. (**A**) Green and roasted coffee extracts. (**B**) Caffeine, caffeic, and chlorogenic acid. Data are represented as mean ± SEM (*n* = 4), with the subtraction of the negative control to delete background noise and the intrinsic compound absorbance at 425 nm. *** represents statistically significant differences with the positive control with *p* < 0.001 and ** *p* < 0.01.

**Figure 4 pharmaceuticals-18-01151-f004:**
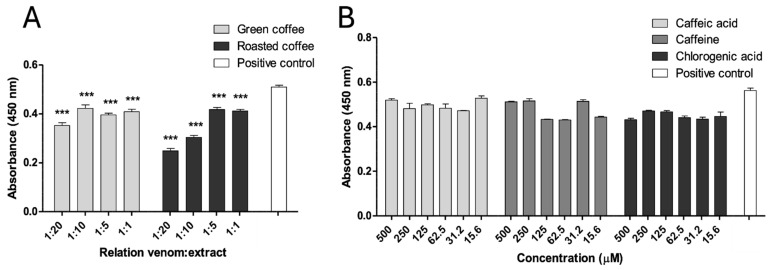
Inhibition of the proteolytic activity of *B. asper* venom. (**A**) Green and roasted coffee extracts. (**B**) Caffeine, caffeic, and chlorogenic acid. Data are represented as mean ± SEM (*n* = 4), with the subtraction of the negative control to delete background noise and the intrinsic compound absorbance at 450 nm. *** represents statistically significant differences with the positive control with *p* < 0.001.

**Figure 5 pharmaceuticals-18-01151-f005:**
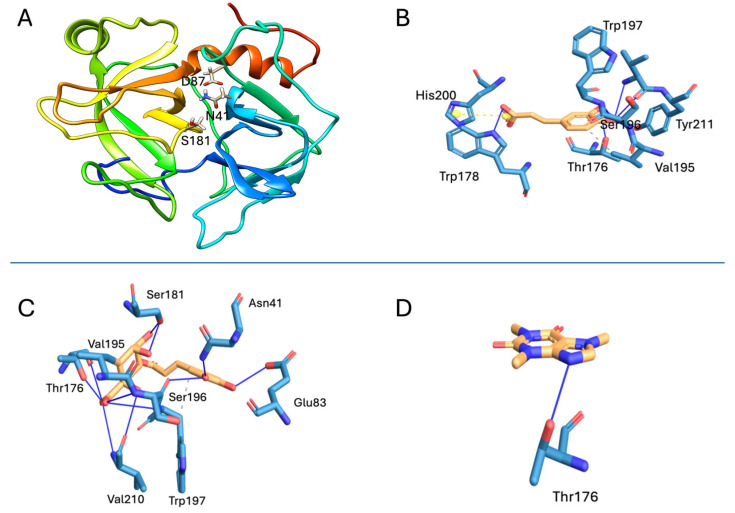
Molecular docking results. (**A**) Modeled structure of the SVSP from *B. asper* venom. The coloring in this view is a N-ter-->C-ter rainbow, with the N terminus being blue and the C terminus red. The catalytic triad of the enzyme is shown in sticks. Interaction of the studied metabolites with the active site residues of the modeled SVSP: (**B**) caffeic acid, (**C**) chlorogenic acid, and (**D**) caffeine. The blue lines represent hydrogen bonds, the yellow lines represent salt bridges, and the gray lines show hydrophobic interactions.

**Figure 6 pharmaceuticals-18-01151-f006:**
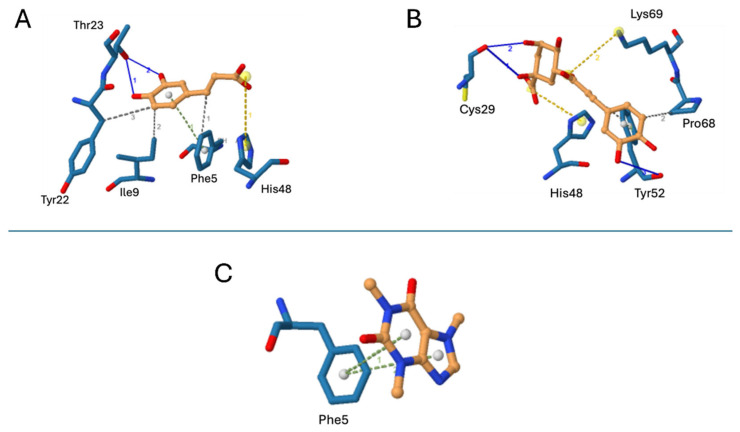
Molecular docking poses with a PLA_2_ (blue sticks) from *B. asper* venom (PDB code 5TFV). Inhibitors are shown in orange sticks. (**A**) Caffeic acid, (**B**) chlorogenic acid, and (**C**) caffeine.The blue lines represent hydrogen bonds, the yellow dotted lines represent salt bridges, the green lines represent π-stacking interactions, and the gray lines indicate hydrophobic interactions.

**Table 1 pharmaceuticals-18-01151-t001:** Weak interactions and affinities between each compound and the modeled *B. asper* serine protease.

Compound	Interactions			Theoretical Affinity (kcal/mol)
	H-Bonds	Salt Bridges	Van der Waals	
Caffeic acid	Thr176, Trp178, Ser196, Trp197, Val210 (2 bonds), Tyr211	His200	Val195	−5.4
Chlorogenic acid	Asn41, Glu83 (2 bonds), Thr176 (2 bonds), Ser181 (2 bonds), Ser196 (2 bonds), Trp197, Val210 (2 bonds)	-	Val195, Trp197	−7.7
Caffeine	Thr176	-	-	−5.1

**Table 2 pharmaceuticals-18-01151-t002:** Weak interactions and affinities between each compound and the *B. asper* phospholipase A_2_ (PDB code 5TFV).

Compound	Weak Interactions			Theoretical Affinity (kcal/mol)
	H-Bonds	Salt Bridges	Van der Waals	
Caffeic acid	Thr23 (2 bonds)	His48	Phe5, Ile9, Tyr22	−5.8
Chlorogenic acid	Tyr52, Cys29 (2 bonds)	His48, Lys69	Pro68, Tyr52	−6.9
Caffeine	-	-	Phe5 (2 bonds)	−5.8

## Data Availability

The original contributions presented in this study are included in this article and the Appendix A. Further inquiries can be directed to the corresponding author.

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
