# Peer review of "Coffea arabica Extracts and Metabolites with Potential Inhibitory Activity of the Major Enzymes in Bothrops asper Venom"

_pharmaceuticals, 2025, doi:10.3390/ph18081151_

Round 1

Reviewer 1 Report

Comments and Suggestions for Authors

The manuscript addresses an important and underexplored area of research: the potential inhibitory effects of Coffea arabica extracts and their metabolites against key enzymatic components of Bothrops asper venom. The study combines biochemical assays with molecular docking, offering a multidimensional view of potential mechanisms of action. However, several serious concerns limit the reliability and interpretability of the findings in its current form.

Major Concerns

  1. The authors report results based on only three replicates (n = 3) per group, which is generally insufficient to draw statistically robust conclusions, particularly in complex venom-inhibition studies. Given the biological variability of both venoms and natural extracts, a higher number of replicates (minimum n = 5) is essential to ensure statistical validity and reproducibility.
  2. Several of the conclusions are drawn from marginal statistical differences, despite large error bars and low replicate counts. For example, in Figure 1, the error in some treatments nearly overlaps with the positive control, casting doubt on the claimed inhibition. The authors are advised to re-analyze the data with caution, report effect sizes or confidence intervals, and moderate language around efficacy claims unless statistical power is improved.
  3. The manuscript states that the coagulation time of the positive control (venom alone) was 20 seconds; however, the visual representation in Figure 1A contradicts this claim, showing much higher values. This mismatch suggests an error in figure labeling, reporting, or data processing that must be corrected before publication.
  4. The manuscript lacks a detailed statistical appendix or data tables summarizing the actual values (mean ± SD or SEM), p-values, and number of replicates per group. Including a supplemental data file or summary table would enhance transparency and enable proper assessment of the findings.
  5. Although the authors mention using a 60:40 isopropanol:water solvent system to extract green and roasted coffee beans, critical parameters are missing. These include: sample mass or volume used, solvent-to-sample ratio, extraction temperature and duration, number of extraction cycles, and yield (e.g., mg extract per gram of coffee). These details are essential to ensure reproducibility and also to assess consistency between batches.

Author Response

We would like to sincerely thank you for your time, effort, and valuable feedback on our manuscript. We greatly appreciate your insightful comments and suggestions, which have helped us to improve the quality and clarity of our work. We have carefully considered each of your observations and have revised the manuscript accordingly. Below, we provide a point-by-point response to all comments, indicating the changes made in the revised version.  

  1. The authors report results based on only three replicates (n = 3) per group, which is generally insufficient to draw statistically robust conclusions, particularly in complex venom-inhibition studies. Given the biological variability of both venoms and natural extracts, a higher number of replicates (minimum n = 5) is essential to ensure statistical validity and reproducibility. 

The amidolytic activity inhibition assays have n=3 replicates per treatment, and the procoagulant, phospholipase A2, and proteolytic activity inhibition assays have n=4 replicates per treatment. Due to the unavailability of reagents, it is currently impossible to repeat the tests to increase the number of replicates. We wish to clarify that the statistical assays were repeated, and new statistical tests were included in the supplementary material to increase confidence in the data. 

2. Several of the conclusions are drawn from marginal statistical differences, despite large error bars and low replicate counts. For example, in Figure 1, the error in some treatments nearly overlaps with the positive control, casting doubt on the claimed inhibition. The authors are advised to re-analyze the data with caution, report effect sizes or confidence intervals, and moderate language around efficacy claims unless statistical power is improved. 

It is important to clarify that for venom procoagulant activity inhibition assays, the metabolite or extract that prolonged the clotting time of the positive control (venom alone) exhibited the most significant inhibitory activity. This is because the venom's serine proteases have a procoagulant effect on plasma. Statistical analyses were repeated, and means, SEM, confidence intervals, and p values were reported in the supplementary material, Tables S1 and S4. 

3. The manuscript states that the coagulation time of the positive control (venom alone) was 20 seconds; however, the visual representation in Figure 1A contradicts this claim, showing much higher values. This mismatch suggests an error in figure labeling, reporting, or data processing that must be corrected before publication. 

The correction was made in line 118.  

4. The manuscript lacks a detailed statistical appendix or data tables summarizing the actual values (mean ± SD or SEM), p-values, and number of replicates per group. Including a supplemental data file or summary table would enhance transparency and enable proper assessment of the findings. 

Statistical analysis with all those suggestions were included as supplemental material, tables S1-S8.  

5. Although the authors mention using a 60:40 isopropanol:water solvent system to extract green and roasted coffee beans, critical parameters are missing. These include: sample mass or volume used, solvent-to-sample ratio, extraction temperature and duration, number of extraction cycles, and yield (e.g., mg extract per gram of coffee). These details are essential to ensure reproducibility and also to assess consistency between batches. 

This information was added in section 4.1 of the methodology, lines 325-337. 

Reviewer 2 Report

Comments and Suggestions for Authors

In this paper, the authors discussed that Arabica coffee bean extracts inhibited the activity of a key enzyme in the venom of Bothrops asper. This is the first report on the effect of coffee bean extracts on the venom of this venomous snake, and is likely to pique readers' interest. However, as a reviewer, I believe that the paper could be better understood if the following points were revised:

1) Regarding the Results: The data shown on lines 106 to 113 of the main text are not included in Figure 1B. The same is true for line 24 of the summary and line 262 of the Discussion, so please correct Figure 1B.

2) Regarding the Results: Please add an explanation in the main text as to why the caffeine, caffeic, and chlorogenic acid preparations listed on line 116 of the legend for Figure 1B were used in this paper.

 3) Regarding the Results: Please explain in the Discussion the authors' thoughts on the fact that chlorogenic acid, listed on lines 124 to 126 of the main text, has a weak inhibitory effect on the amidolytic activity of snake venom.

4) Regarding the Results: As with 3), please explain in the Discussion the authors' thoughts on the result, listed on lines 149 to 150 of the main text, that caffeine, caffeic, and chlorogenic acid preparations do not inhibit proteolytic activity.

 5) Regarding the discussion: Please include the limitations of the data presented in this paper in the discussion.

Author Response

We would like to sincerely thank you for your time, effort, and valuable feedback on our manuscript. We greatly appreciate your insightful comments and suggestions, which have helped us to improve the quality and clarity of our work. We have carefully considered each of your observations and have revised the manuscript accordingly. Below, we provide a point-by-point response to all comments, indicating the changes made in the revised version. 

In this paper, the authors discussed that Arabica coffee bean extracts inhibited the activity of a key enzyme in the venom of Bothrops asper. This is the first report on the effect of coffee bean extracts on the venom of this venomous snake, and is likely to pique readers' interest. However, as a reviewer, I believe that the paper could be better understood if the following points were revised: 

  1. Regarding the Results: The data shown on lines 106 to 113 of the main text are not included in Figure 1B. The same is true for line 24 of the summary and line 262 of the Discussion, so please correct Figure 1B. 

Figure 1B was corrected. We apologize for the mistake.  

2) Regarding the Results: Please add an explanation in the main text as to why the caffeine, caffeic, and chlorogenic acid preparations listed on line 116 of the legend for Figure 1B were used in this paper. 

This information was added in lines 107-112.  

 3) Regarding the Results: Please explain in the Discussion the authors' thoughts on the fact that chlorogenic acid, listed on lines 124 to 126 of the main text, has a weak inhibitory effect on the amidolytic activity of snake venom. 

We certainly do not know why the individual compounds inhibit differentially the amidolytic and coagulant activities of the snake venom. Nevertheless, we add some information (lines 286-293) that may contribute to the explanation of the obtained result.  

4) Regarding the Results: As with 3), please explain in the Discussion the authors' thoughts on the result, listed on lines 149 to 150 of the main text, that caffeine, caffeic, and chlorogenic acid preparations do not inhibit proteolytic activity. 

We certainly do not know why the individual compounds do not inhibit the proteolytic activity of the snake venom. We hypothesize that at the concentrations used the inhibition by the extracts may be due to the presence of other molecules or through synergistic effects between them. This information was added in lines 253 and 257.  

 5) Regarding the discussion: Please include the limitations of the data presented in this paper in the discussion. 

The limitations of the data were included in lines 308-312 

Once again, thank you for your constructive input and your contribution to enhancing our work.